# Explaining Low Dimensional Representation, a reproduction

## Reproducibility Summary

**Scope of Reproducibility**

This report covers our reproduction of the paper 'Explaining Low dimensional Representation' [8] by Plumb et al. In this paper, a method (Transitive Global Translations, TGT) is proposed for explaining different clusters in low dimensional representations of high dimensional data. They show their method outperforms the Difference Between the Means (DBM) method, is consistent in explaining differences with few features and matches real patterns in data. We verify these claims by reproducing their experiments and testing their method on new data. We also investigate the use of more complex transformations to explain differences between clusters.

**Methodology**

We reproduce the original experiments using their source code. We also replicate their findings by re-implementing the authors' method in PyTorch [7] and evaluating on two of the dataset used in the paper and two new ones. Furthermore, we compare TGT with our own extension of TGT, which uses a larger class of transformations.

**Results**

We were able to reproduce their results using their code, yielding mostly similar results. TGT generally outperforms DBM, especially when explanations use few features. TGT is consistent in terms of the features to which it attributes cluster differences, across different sparsity levels. TGT matches real patterns in data. When extending the types of functions used for explanations, performance did not improve significantly, suggesting translations make for adequate explanations. However, the scaling extension shows promising performance on the modified synthetic data to recover the original signal.

**What was easy**

The easiest part was running the existing code with the pre-trained model files. The original authors had set up their code base in an organized manner with clear instructions.

**What was difficult**

The first difficulty that we encounter was finding the right environment. The source code depends on deprecated functionality. The clustering method they used, had to be re-implemented for us to use it in our replication. Another difficulty was the selection of clusters. The authors did not prove a consistent method for selecting clusters in a latent space representation. When retraining the provided models, we get a latent space representation different to the original experiments. The clusters have to be manually selected. The metrics that they used to evaluate their explanations are also depend on the clustering. This means that there is some variability in the exact verification of reproducibility.

**Communication with original authors**

We asked the original authors for clarification on how to choose the $\epsilon$ hyper-parameter. However, it became apparent that we had misread, and the procedure is indeed adequately reported in the paper.

## 1 Introduction

The *curse of dimensionality* [3] is a long-standing problem in Machine Learning. Data in many domains and applications (e.g. Bioinformatics) has high-dimensional representations. Finding patterns in such high-dimensional data is a challenging task. To this end, *dimensionality reduction* [12] techniques have greatly helped in data-analysis, information extraction, building computational models, and in doing inference. Given an input $x \in \mathbb{R}^d$, dimensionality reduction learns a function $r : x \mapsto r(x)$, $r(x) \in \mathbb{R}^m$, where $m << d$. Such a dimensionality reduction function $r$ naturally arises in deep learning due to the expressivity and representational power of neural networks. The goal of $r$ is to encode useful knowledge about the input space, thus providing distinctive information in the transformed output $r(x)$. This results in "clusters" or "groups of points" in the transformation space. The downside of this exercise, however, is that the output space is usually non-interpretable. There is usually no easy way to know what information is present in the transformed points $r(x)$ and what sort of distinctive knowledge they contain.

In this work, we reproduce the paper 'Explaining Groups of Points in Low-Dimensional Representations' by Plumb et al. [8]. This paper proposes a method for explaining different clusters in latent space representation. They look at the problem of explaining the points in the latent space representation through the lens of Interpretability in Machine Learning. We reproduce their findings and expand upon their work with our an extension. We extend their research by applying their method to a larger class of explanation functions and testing their method on new dataset. We further investigate the efficacy of the explanations using a probing classifier [2].

## 2 Methodology

Counterfactual Explanations [11] have emerged as an active research area in the field of Interpretable Machine Learning. A counterfactual explanation is defined as the smallest perturbation to the input that would change the output of a machine learning model. As such, these explanations are promising as they can provide suggestive recourse to the beneficiary in a machine learning based decision system. As an interpretable machine learning problem, Plumb et al. [8] aim to find such counterfactual explanations in order to explain the differences between the groups in latent space. To this end, they employ the function $r$ itself to find what perturbation $\delta$ needs to be made to the input $x \in \mathbb{R}^d$ so that $r(x + \delta)$ belongs to the different target group. The goal is to find the *global* explanations that apply to the whole group as opposed to the *local* explanations which explain only individual examples [4]. Furthermore, the explanations need to be *sparse* for them to be interpretable by practitioners. Finally, these explanations should be be both *symmetric* and *transitive*. To obtain these *Global Counterfactual Explanations*(GCE), the authors propose the algorithm called, Transitive Global Translations (TGT), explained hereafter.

Following the previous notation, let $r : \mathbb{R}^d \to \mathbb{R}^m$ denote our dimensionality reduction function, where $d$ is the dimensionality of the input space and $m$ is the latent space's dimensionality. Suppose $X_i, X_j \subset \mathbb{R}^d$ get mapped to the clusters $R_i, R_j \subset \mathbb{R}^m$ respectively. The goal is to define the transformation $t_{i \to j} : \mathbb{R}^d \to \mathbb{R}^d$ on $x \in X_i$ as $x^{'} = t_{i \to j}(x)$, so that $r(x^{'}) \in R_j$, or equivalently $x^{'} \in X_j$.

The proposed algorithm TGT considers the transformations of the form $t_{i \to j}(x) = x + \delta_{i \to j}$. To find the optimal parameters of the transformation function, authors imply a compressed-sensing based objective function as below:

$$l(\delta_{i \to j}) = \|r(t_{i \to j}(\bar{X}_i)) - \bar{R}_j\|_2^2 + \lambda\|\delta_{i \to j}\|_1 \tag{1}$$

where $\lambda\|\delta_{i \to j}\|_1$ is a regularization term to incentivize sparser explanations, and $\bar{X}_i \in \mathbb{R}^d$ and $\bar{R}_j \in \mathbb{R}^m$ denote the means of the clusters in the input space and latent space respectively. Given clusters $0, 1, \ldots, n$, we get a total of $\frac{1}{2}n(n + 1)$ transformations. To further increase sparsity, we can truncate $\delta_{i \to j}$ to only the $k$ features with the largest absolute value, for some $k$. An issue with this is that the translation using the truncated $\delta_{i \to j}$ might no longer correctly transform inputs that get mapped to $R_i$ into inputs that get mapped to $R_j$.

Furthermore, the transformations $t_{i \to j}$ have to adhere to several mathematical properties. Namely, for any clusters $i, j, k$ these transformations should be : a) Symmetric, i.e. $t_{i \to j} = t_{j \to i}{}^{-1}$ and b) Transitive, i.e. $t_{j \to k} \circ t_{i \to j} = t_{i \to k}$. From these properties it follows that $t_{i \to i}$ is the identity function $\mathcal{I}$ as

$$t_{i \to i} = t_{i \to 0} \circ t_{0 \to i} = t_{i \to 0} \circ t_{i \to 0}{}^{-1} = \mathcal{I} \tag{2}$$

We define this condition as *self-similarity*. Furthermore, the group of translations is uniquely defined by $t_{0 \to 1}, \ldots, t_{0 \to n}$, because for any $i, j$:

$$t_{i \to j} = t_{0 \to j} \circ t_{i \to 0} = t_{0 \to j} \circ t_{0 \to i}{}^{-1} \tag{3}$$

Plumb et al. [8] compare their method against the naive baseline of Difference Between the Means (DBM). With DBM, each transformation is still a translation: $t_{i \to j}(x) = x + \delta_{i \to j}$. However, now $\delta_{i \to j} = (\bar{X}_j - \bar{X}_i)$. We also use this as a baseline for comparison in this report.

Since translations are a very narrow class of functions, we expanded upon the research by investigating other transformations that still satisfy the GCE requirements. We investigate the transformations of the form $t_{0 \to i}(x) = \exp(\gamma_{0 \to i}) \odot x + \delta_{0 \to i}$. These always have a well defined inverse, given by $t_{0 \to i}^{-1}(x) = \exp(-\gamma_{0 \to i}) \odot (x - \delta_{0 \to i})$ and only have $\mathcal{O}(d)$ parameters. The inclusion of scaling could enhance performance, while the necessary components of GCE are maintained.

## 2.1 Metrics to evaluate Global Counterfactual Explanations

To measure the efficacy of the transformation function $t_{i \to j}$, the authors propose two metrics, *Coverage* and Correctness.

1. The **Coverage** $(cv(t_{i \to j}))$ is the fraction of points $a \in R_j$ for which there is a point $b \in X_i$ such that $\|r(t_{i \to j}(b)) - a\|_2 < \epsilon$, i.e.

$$cv(t_{i \to j}) = \frac{1}{|R_j|} \sum_{a \in R_j} \mathbb{I}\left[\exists b \in X_i | \|r(t_{i \to j}(b)) - a\|_2 < \epsilon\right] \tag{4}$$

2. The **Correctness** $(cr(t_{i \to j}))$ is the fraction of points $b \in X_i$ for which there is some $a \in R_j$ such that $\|r(t_{i \to j}(b)) - a\|_2 < \epsilon$, i.e.

$$cr(t_{i \to j}) = \frac{1}{|X_i|} \sum_{a \in R_j} \mathbb{I}\left[\exists a \in R_j | \|r(t_{i \to j}(b)) - a\|_2 < \epsilon\right] \tag{5}$$

Note that both these metrics have the hyperparameter $\epsilon$ which is to be chosen carefully. When $i = j$ we do not count the point itself, there must be some other point within distance $\epsilon$. [1]

Furthermore, the **Similarity** metric measures the consistency of the explanations at different sparsity levels. Given two explanations $e_1, e_2$ where $e_1$ is more sparse than $e_2$, the similarity of $e_1$ and $e_2$ is defined as

$$sim(e_1, e_2) = \frac{\sum_i |e_1[i]| \mathbb{1}(e_2[i] \neq 0)}{\|e_1\|_1} \tag{6}$$

This is equal to $1$ if $e_1$ uses a subset of the features that $e_2$ uses. By definition, DBM always has similarity $1$.

# 3 Scope of Reproducibility

We investigate the following claims from the original paper:

1. In terms of the average correctness and coverage, TGT performs equally well or better than the DBM method. This remains true, especially for sparser explanations.

2. TGT explanations have similarity close to $1$. It is consistent in which features it uses for explanations across different sparsities.

3. TGT correctly identifies known causal structure in data.

4. Furthermore, TGT explanations are consistent. When altering the dataset by adding a copy of a cluster with a specific feature altered, TGT recovers the modification with little change to the other explanations.

# 4 Methodology of Reproducibility

We make use of the code made available by the original authors [2] for our pilot investigative study. We first verify that the provided models and explanations stay true to the claims made in the paper. We further retrain their models on the provided dataset. We also made our own PyTorch [7] implementation to to further verify the claims, and to perform experiments with the proposed extension.

---

[1] We use this definition to set the value for epsilon, as explained in the *Methodology* section of the original Paper.
[2] https://github.com/GDPlumb/ELDR

## 4.1 Model description

We identify that the scope of the original paper is to explain clusters in the low-dimensional representations. However, obtaining meaningful and discernible low-dimensional representations is an active area of research. The original authors employ a t-SNE [10] objective based Variational Autoencoder (called, henceforth, as scVIS) [5] as the $r$ function. They make use of library[3] by the original scVIS authors in their implementation. We also implement this model in Pytorch for our experiments. However, we deliberately decide not to match the model implementation exactly. This is done to study the model-agnosticism of the TGT algorithm. By design, TGT should be able to explain the clusters for any differentiable $r$ function. However, we maintain that $r$ should give discernible latent representations with preserved global structure in the data. In our implementation of the scVIS library, we therefore do not employ the hyperparameters and the training settings from the original library.

## 4.2 Dataset Description

We reproduce the findings of the authors on four datasets that they used. We use two of these datasets as well as two new ones to test our PyTorch implementation.

1. **Single cell RNA** [9]: This dataset has 13166 features. We use the same number of clusters at the original authors, 18 in this case.

2. **UCI Boston housing** This dataset has 506 entries with 13 features. We use 6 clusters for both reproduction and replication.

3. **UCI Heart disease** This dataset has 303 entries with 13 features and 1 binary label. We used 8 clusters in the reproduction and 4 in the replication. The data was normalized to be in the range [0, 1].

4. **UCI Iris** This dataset has 150 entries with 4 features and 1 ternary label. Ran in the reproduction with 3 clusters. N = 150

5. **Breast Cancer Wisconsin (Diagnostic)** [4] This dataset has 569 entries with 30 features and 1 binary label. We use 3 clusters in the replication.

6. **Pima Indians Diabetes Database** [5] This dataset has 768 entries with 8 features and 1 binary label. We used 3 clusters in the replication. The data was normalized to be in the range [0, 1].

Note that the number of clusters depend on the latent-space representation, and thus, are user dependent.

## 4.3 Hyperparameters

**Tensorflow [1] Experiments** For the reproduction of the original experiments, we use the same hyperparameters as the original authors.

**Pytorch** For our implementation of the scVIS model, we use l2 regularization of 0.001, learning rate 0.01, and perplexity of 10. Furthermore, the degree-of-freedom for the studentT distribution is set to 2.0. Perplexity and the degree-of-freedom is used same as the original scVIS implementation. We use validation set to monitor the training process of the scVIS model, and stop training when the ELBO(Evidence Lower BOund)[6] stops improving. For training the TGT explanations, we closely follow the settings from Plumb et al. [8]. We initialize the deltas$_s$ as zero vectors. We tune the regularization parameter $\lambda$ by grid search over a fixed range $[0.0, 5.0]$ incremented by 0.5. Defining the metrics for TGT requires careful setting of the $\epsilon$ hyper-parameter. We follow the *self-similarity* condition (transformations of clusters to themselves should, theoretically, have correctness and coverage to be 1.0), and increase the $\epsilon$ in the range $[0.0, 2.0]$ with increments of 0.02 until the correctness and coverage metrics are greater than 0.95. Furthermore, we use the truncation values(TV)(refer Table 1) to evaluate on the sparsity of the explanations. For the Pima Indians Diabetes Database and Breast Cancer Wisconsin(Diagnostic) dataset, we use the same truncation values as for UCI Boston Housing dataset.

## 4.4 Experimental setup and code

We closely follow the experimental setup in the original paper for our experiments. We make our Pytorch code available [6] to further support the reproducible research. We reran the code of the original authors with new clustering models

---

[3]`https://github.com/shahcompbio/scvis`

[4]https://www.kaggle.com/uciml/breast-cancer-wisconsin-data

[5]https://www.kaggle.com/uciml/pima-indians-diabetes-database

[6]https://github.com/elfrink1/FACT

| Dataset | Truncation Values(TV) | $\epsilon$ |
|---|---|---|
| Single Cell RNA | 50, 100, 250, 500, 1000, 15000 | 0.75 |
| Heart Disease | 1, 3, 5, 7, 9, 11, 13 | 1.0 |
| Housing | 1, 3, 5, 7, 9, 11, 13 | 1.5 |
| Iris | 1, 2, 3, 4 | 0.75 |

Table 1: Truncation Values (TV) and $\epsilon$ value used for each of the dataset.

| Explanation | $x_1$ | $x_2$ | $x_3$ | $x_4$ |
|---|---|---|---|---|
| $0 \rightarrow 1$ | -1.01 | -0.02 | 0.00 | -0.88 |
| $0 \rightarrow 1$ | -1.05 | 0.99 | 0.00 | -0.88 |
| $0 \rightarrow 3$ | 0.00 | 0.89 | 0.00 | 0.00 |

Table 2: Explanations for the synthetic dataset as given by our implementation. Note that both DBM and TGT are able to infer that the $x_3$ is not causing any cluster. However, the authors' claim that TGT also discovers that $x_4$ doesn't cause any cluster cannot be verified.

and new explanations. We optimize the compressed-sensing based objective function for the TGT algorithm using the gradient descent algorithm. Our scaling extension is easily integrated in the source code, and can be optimized in a similar way. We train the scVIS models on the Lisa computing cluster [7]. We use approximately 30 hours of GPU time. We train the TGT explanations on CPU (Intel i5).

# 5 Results

For the reproduction of the authors' experiments, we achieve approximately similar results to the original paper. The TGT method does seem to outperform DBM method. The TGT explanations also have high similarity across sparsity levels. However, the TGT algorithm is unable to identify known causal structure in synthetic data with as good precision as reported in the original paper. We are also unable to match the results on the modified and corrupted data to a good precision. We describe the results in the following sections:

## 5.1 Results reproducing original paper

### 5.1.1 Coverage, Correctness and similarity

In figure 1, we can see a comparison between the correctness, coverage and similarity of the TGT and DBM methods. Note that the DBM always has similarity 1. The similarity of TGT stays between 1 and 0.9, which supports claim 2.

We see that the coverage and correctness are similar for the UCI Heart disease dataset. On the UCI Iris dataset, the coverage is comparable but the correctness is better for TGT. In both housing and RNA, the coverage and correctness are better at less features and similar for more features. Overall, these results support claim 1, especially for a small amount of features.

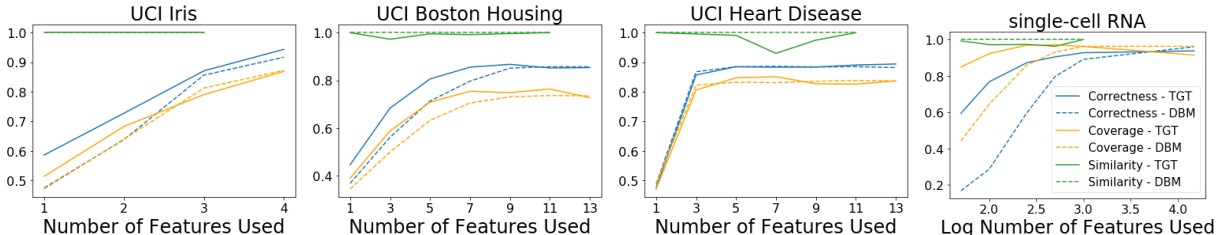

Figure 1: Comparison of the metrics(Correctness, Coverage, and Similarity) across different datasets for reproduction experiments.

---

[7]https://userinfo.surfsara.nl/systems/lisa

## 5.2 Explaining Causal Structure in the Synthetic Data

We verify the claim that TGT identifies the causal structure in the data (claim 3). The synthetic dataset is generated same as the original paper, i.e. $x_1, x_2 \sim \mathcal{N}(0, 0.2) + \text{Bern}(0.5)$, $x_2 \sim \mathcal{N}(0, 0.05)$, $x_4 \sim x_1 + \mathcal{N}(0, 0.05)$. Note that this dataset has four different clusters, caused by the first two dimensions $x_1$ and $x_2$, $x_3$ is noise, and $x_4$ is correlated with $x_1$ and $x_2$. The authors claim that for this synthetic data, TGT is able to find that $x_4$ is not the cause for any group. However, the said claim cannot be re-verified. Interestingly, the re-run of their code doesn't provide the justification either to the degree as mentioned in the paper. We observe that both TGT and DBM are able to identify $x_3$ is not causing any groups. Thus, in this scenario, both TGT and DBM are comparable. Refer table 2 for the explanations obtained. We, hereby note, that the explanations vary across multiple runs and we use the experimental setup same as the original authors. However, the values across the third dimension are consistently approximately 0.

### 5.2.1 Feature modifications

For each of the UCI datasets, the original authors add a 'corrupted' version where an extra cluster is added with artificial feature modification. With the exception of the modified features, the corrupted class is a copy of a chosen target class. They train TGT explanations using both the original scVIS model for the respective dataset and a model retrained on the corrupted dataset. We reproduce these experiments to see if TGT correctly attributes the difference between the target and corrupted class to the right features. Refer to Appendix A.1 for the illustrated figures and description. Overall, we observe that TGT is unable to identify the modifications to as good a precision as reported in the original paper. TGT is able to identify the modification for the UCI Iris dataset. For UCI Heart Disease Dataset (figure 7), it does not identify the features modified and on the UCI Boston Housing Dataset (figure 6), it identifies noisy modifications. However, with the retrained scVIS model and new representations, TGT is consistent in identifying the modifications across all the datasets.

## 5.3 Results beyond original paper

### 5.3.1 PyTorch replication

We also replicate the TGT algorithm in PyTorch. Our Pytorch implementation includes the entire method along with the scVIS clustering method. In our implementation, we use the Scikit learn [8] kmeans module for our cluster selection as opposed to the manual clustering in the Tensorflow implementation. However, our number of clusters argument to the kmeans algorithm was informed by the learned low-dimensional representations for each dataset. Due to differences in the clustering model and cluster selection, we cannot directly compare the coverage and correctness metrics between our Pytorch replication and the TensorFlow reproduction. We additionally experiment with our scaling extension to the TGT algorithm. In the scaling extension of the TGT algorithm, along with the $\delta$ ($\delta$) parameters, each cluster now has a $\gamma$ ($\gamma$) parameters. The transformation from cluster 0 to $i$ is now given by: $t_{0 \to i} = e^{\gamma_i} \odot x + \delta_i$ The gammas($\gamma_s$) are truncated just like the deltas and their $L1$ norm is added to the regularization term. Note that these transformations are strictly more expressive. If $\gamma$ is the zero vector, these transformations reduce to regular TGT.

### 5.3.2 UCI Heart Disease and UCI Boston Housing Dataset

In figure 2 we see the results of our replication on the UCI Boston Housing and UCI Heart Disease dataset. For the UCI Boston Housing data, the TGT method seems to slightly outperform DBM both with and without scaling. This supports claim 1. The deltas($\delta_s$) and gammas($\gamma_s$) show high similarity, supporting claim 2. For the UCI Heart Disease dataset, we do not see a difference in performance without scaling while TGT with scaling performs slightly worse.

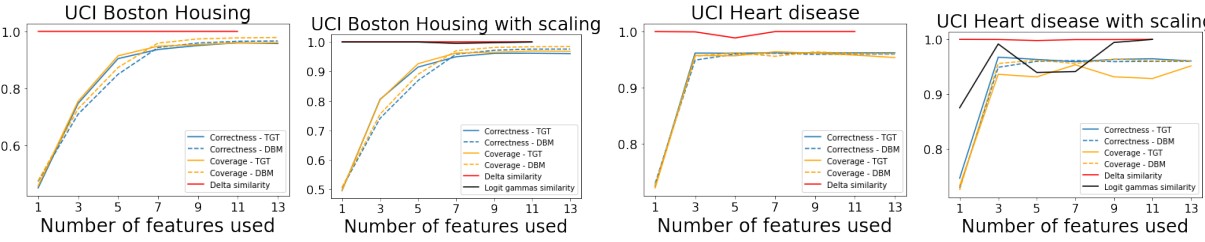

Figure 2: Results for the PyTorch replication for UCI Boston Housing and UCI Heart Disease dataset.

[8]https://scikit-learn.org/stable/index.html

| | 0 to 1 | | 1 to 0 | | $\delta_1$ | $\delta_2$ | $\delta_3$ | $\delta_4$ | $\gamma_1$ | $\gamma_2$ | $\gamma_3$ | $\gamma_4$ |
|---|---|---|---|---|---|---|---|---|---|---|---|---|
| | $Cr$ | $Cv$ | $Cr$ | $Cv$ | | | | | | | | |
| **TGT** | 1 | 0.529 | 0.333 | 1.000 | 2.581 | 0.014 | 0.001 | 0.842 | - | - | - | - |
| **Scaling** | 1 | 0.529 | 0.520 | 1.000 | 2.608 | 0.023 | -0.001 | 0.927 | 0.879 | 0.009 | 0.002 | 0.007 |

Table 3: The deltas($\delta_s$) and gammas($\gamma_s$) for the mapping from group 0 to group 1 on the modified synthetic dataset for regular TGT and TGT with scaling. *Cr* and *Cv* indicate correctness and coverage, respectively.

### 5.3.3 Breast Cancer Wisconsin (Diagnostic) and Pima Indians Diabetes Database

In figure 3 we see the results for the Pima Indians Diabetes and Breast Cancer Wisconsin (Diagnostic) dataset. For the diabetes dataset, TGT with and without scaling outperforms DBM when more than one feature is used. This supports claim 1. Since the delta ($\delta$) similarity is close to 1, claim 2 is also supported. For the Breast Cancer dataset, we see similar performance for DBM and TGT and slightly worse performance with scaling. The deltas ($\delta_s$) still have high similarity, supporting claim 2.

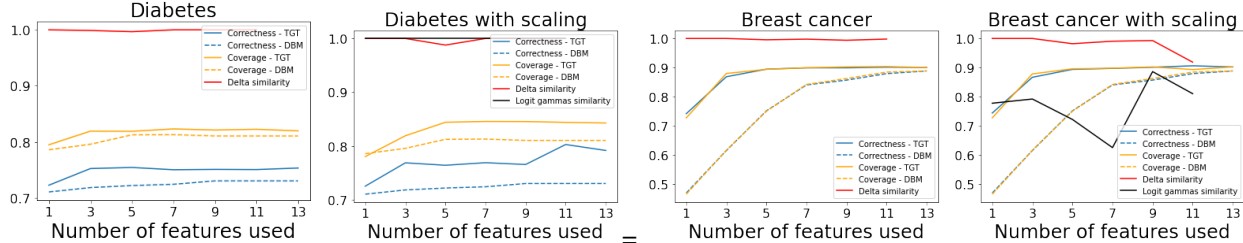

Figure 3: Results of the PyTorch replication on PIMA Indians Diabetes and Breast Cancer Wisconsin (Diagnostic) Dataset.

### 5.3.4 Scaling extension

In figure 2, we can see the difference in performance on two dataset included in the original experiments. Scaling does not seem to improve performance on the UCI Boston Housing dataset and slightly decreases performance on the UCI Heart Disease dataset. The similarity of the gammas ($\gamma_s$) is mostly above 0.9.

In figure 3, we see the same metrics for the Breast Cancer and Pima Indians Diabetes Dataset. For the Diabetes dataset, the performance improves slightly and the gammas($\gamma_s$) show high similarity. For the Breast Cancer dataset, the performance is about the same but the gammas($\gamma_s$) show relatively low similarity.

Altogether, these results suggest that the addition of scaling does not significantly improve the accuracy and correctness while making the transformations more complex. Based on our experiments, we do not recommend the addition of scaling in the explanation functions, and conclude that the original TGT is expressive enough.

### 5.3.5 Experiment with Modified Synthetic Data

In order to study the efficacy of the proposed scaling function, we perform experiments on the synthetic dataset. We modify one of the groups of points $G$ by performing the operation $ax_i^k + b$, where $i$ corresponds to the group number and $k$ denotes the feature dimension which we modify. We define $a \sim \mathcal{U}(1.0, 2.0)$ and $b \sim \mathcal{U}(-0.5, 1.0)$. We add modified group $G'$ into the original data $D$ to get the new data $D'$. We follow the experiment setup from the original paper as: a) $r(G')$ should form a different group of its own. b) $G'$ should be within the distribution of the original $D$. In this study, we want to investigate whether the TGT with scaling is able to recover the modifications, and if in doing so it affects the explanations between other groups. The sampling procedure gave a=2.0, b=0.60 and we keep k=0. We observe that the explanations with scaling are able to recover the modification to an approximate degree(scaling factor $e^\gamma \approx 2.38$, actual a=2.0), and give better correctness as compared to the regular TGT (refer figure 3). Interestingly, the translations explanations of the scaling extension are approximately equal to the deltas of the regular TGT. The exact results can be found in Table 3. Figures 8 and 9 in the appendix show the data spread and resulting translations.

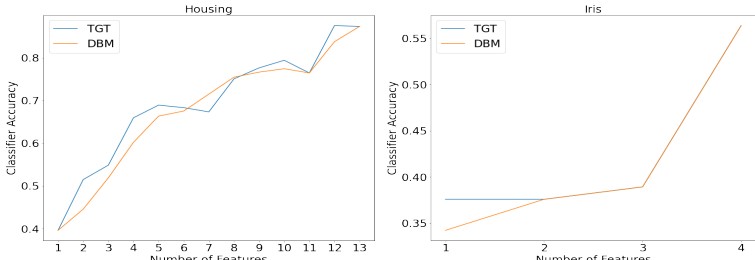

Figure 4: Classification accuracy of probing classifier at different sparsity levels for Housing (left) and Iris (right) dataset.

### 5.3.6 Experiments with Probing Classifier

To further investigate the efficacy of TGT explanations, we use a probing-classifier [2] as a proxy to study the qualitative differences of the features selected by TGT and DBM. For each cluster, we train a binary classifier with features ranked highest by TGT and DBM at different sparsity levels K. We compute the overall accuracy at each sparsity level using the ensemble of these binary classifiers. As can be seen in Figure 4, the results demonstrate that for sparser explanations, TGT selects features that lead to higher accuracy of the ensemble classifier than those selected by DBM. This further validates the paper's claim that TGT leads to better sparse explanations as compared to DBM. Furthermore, we also use the probing classifier to understand the differences between the groups. For each pair of group, we train a Binary Linear Classifier to predict the group of a test point. We, then, investigate the feature importances of the classifier towards decision making. We ascertain that the features classifier give more importance to while decision making are the defining property of the class. Interestingly, we find that the more important features according to the classifier correspond to the explanations provided by the TGT algorithm. Refer to figure 11. This provides further evidence that TGT is able to find real distinctive signals as explanations.

## 6 Discussion

Based on the reproduction of the original experiments, claims 1 and 2 seem to hold, the experiments for claim 4 do not all support it, but the claim does seem to hold. Claim 1 and 2 seem to hold in particular for sparse explanations. The evidence for claim 3 is inconclusive. The coverage and correctness in our reproduction were not always the same as in the original paper. It is difficult to compare these metrics for different clustering outcomes, as they depend on the $\epsilon$ parameter which depends on the clustering.

A major difficulty in reproduction is the cluster selection. When retraining the scVIS model, the latent space representation structure changes. The authors provide no method as to determine the different clusters other than visual inspection. Cluster selection could be an explanation for the differences in results between the original experiments and our reproduction. To verify the results with more confidence, a robust method for cluster selection might be required.

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

# A Extra figures

## A.1 Experiments on Corrupted datasets

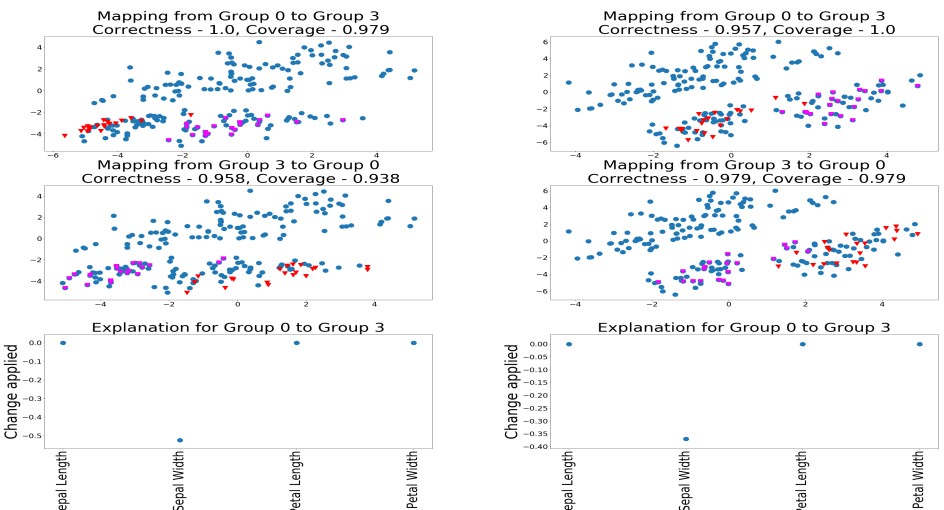

Figure 5: Explanation for corrupted features on UCI Iris dataset. Feature modified is 1(Sepal Width). Left: Visualization of the TGT explanations on the modified dataset. Right: Visualization of the TGT explanations with scVIS retrained on the modified dataset. We observe that the TGT explanations are robust to the modifications.

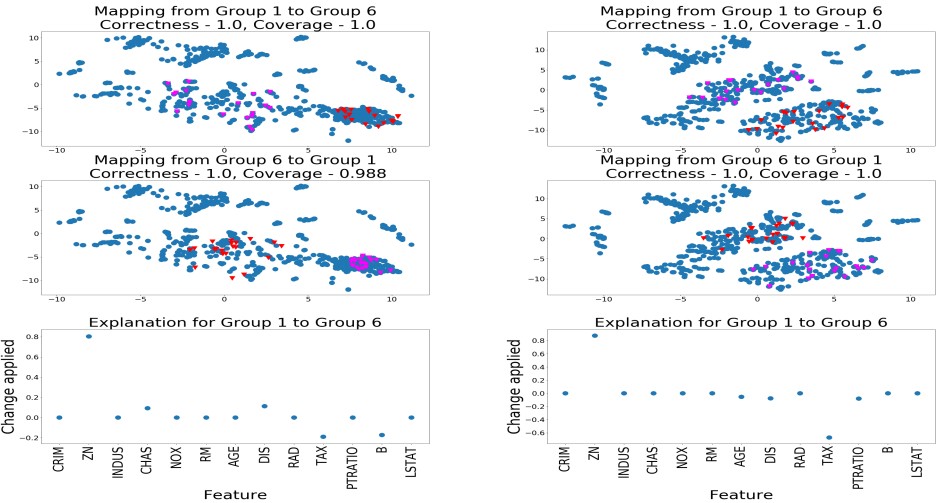

Figure 6: Explanation for corrupted features on UCI Boston Housing dataset. We modify the features 1(ZN) and 9(TAX). Left: Visualization of the TGT explanations on the modified dataset. We observe that TGT returns noisy explanations in this case. Right: Visualization of the TGT explanations with scVIS retrained on the modified dataset. With retrained scVIS model, TGT is able to recover the modifications.

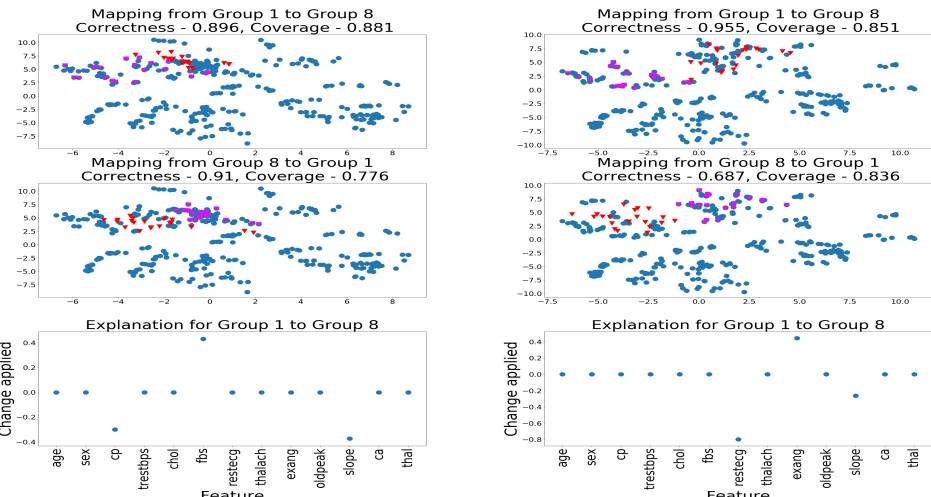

Figure 7: Explanation for corrupted features on UCI Heart Disease dataset. Left: Visualization of the TGT explanations on the modified dataset. We modified the 6(restecg) and 8(exang). However, the TGT recovers modifications in features 2(cp), 5(fbs), and 10(slope) instead. Right: Visualization of the TGT explanations with scVIS retrained on the modified dataset. With retrained scVIS model, TGT recovers the modified features along with 10(slope) feature. This observation does not entirely support the claim 4.

## A.2 Synthetic data

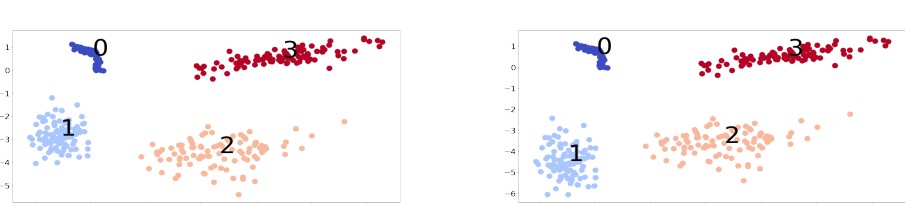

Figure 8: a) Synthetic data b) Synthetic data with the modification applied. We modify the data from group 1 across the $0^{th}$ dimension by $ax_1^0 + b$. Here a and b are 2.0, 0.60 respectively.

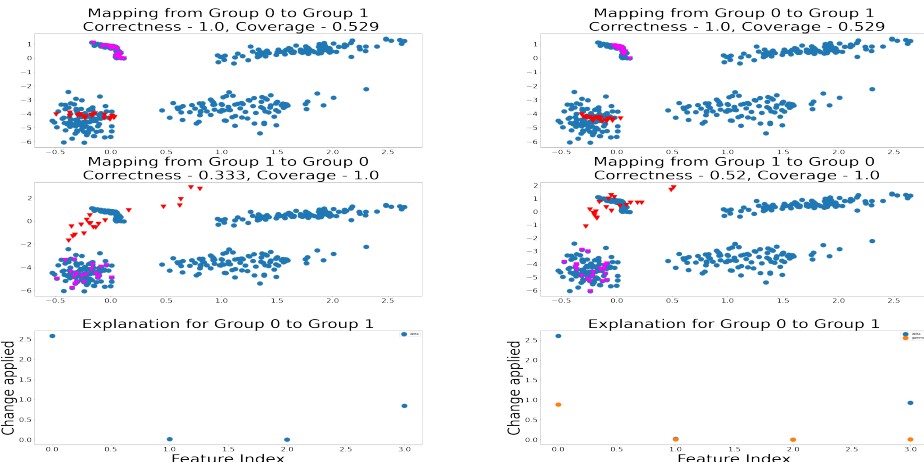

Figure 9: We compare the explanations from the TGT algorithm (left) and the TGT with scaling extension algorithm(right) on the modified synthetic data. We can observe that the TGT with scaling extension has better correctness, and is able to identify the scaling we have applied across the first dimension (i.e. k=0). The $\gamma$ for this dimension is 0.87, which means the scaling factor is $e^{\gamma} \approx 2.38$. Moreover, the translation parameters are approximately same in both the variants of the TGT.

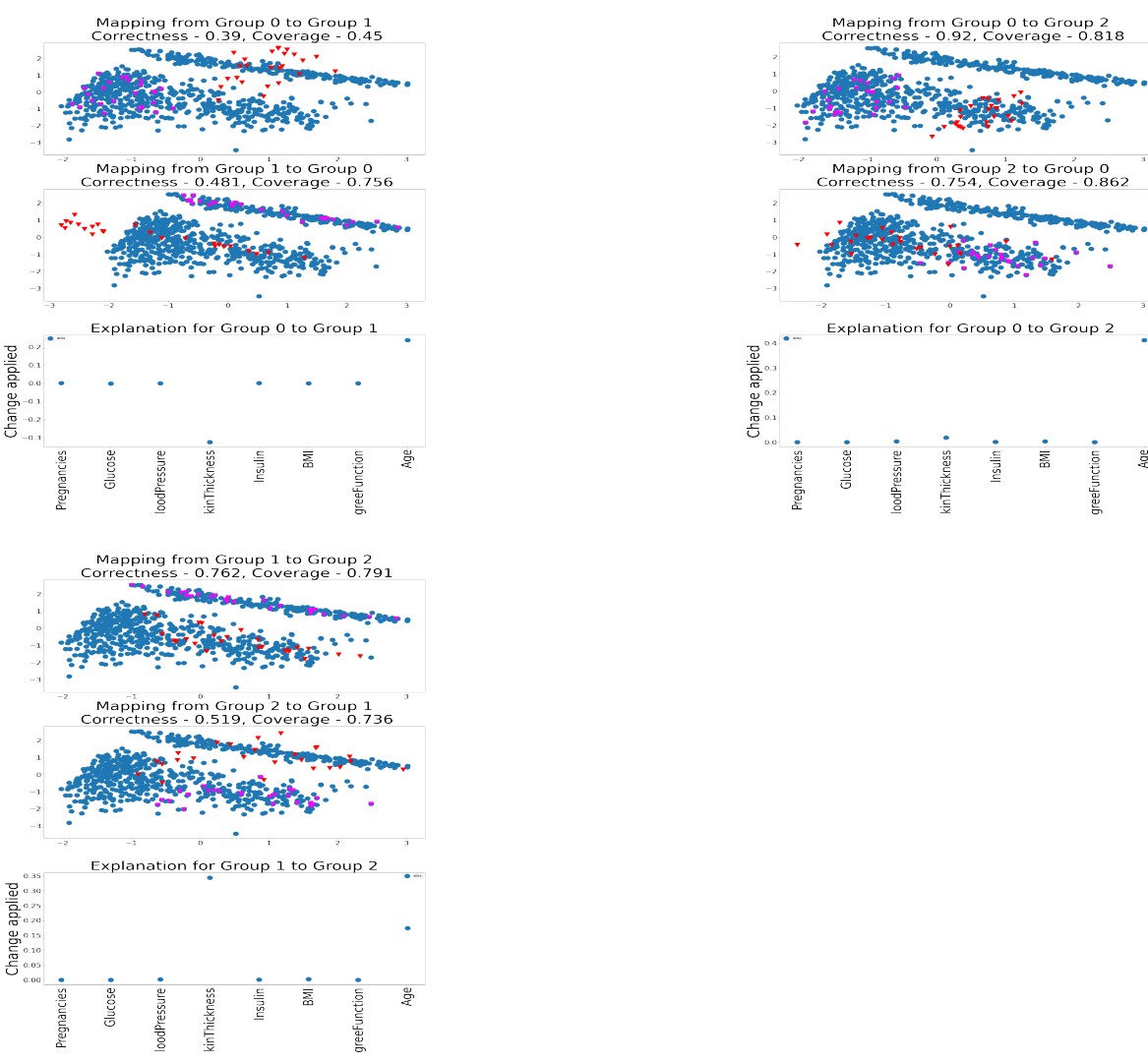

Figure 10: Explanations between different groups for the Pima Indians Diabetes Database.

 ## A.3  Probing Classifier and Feature Importance

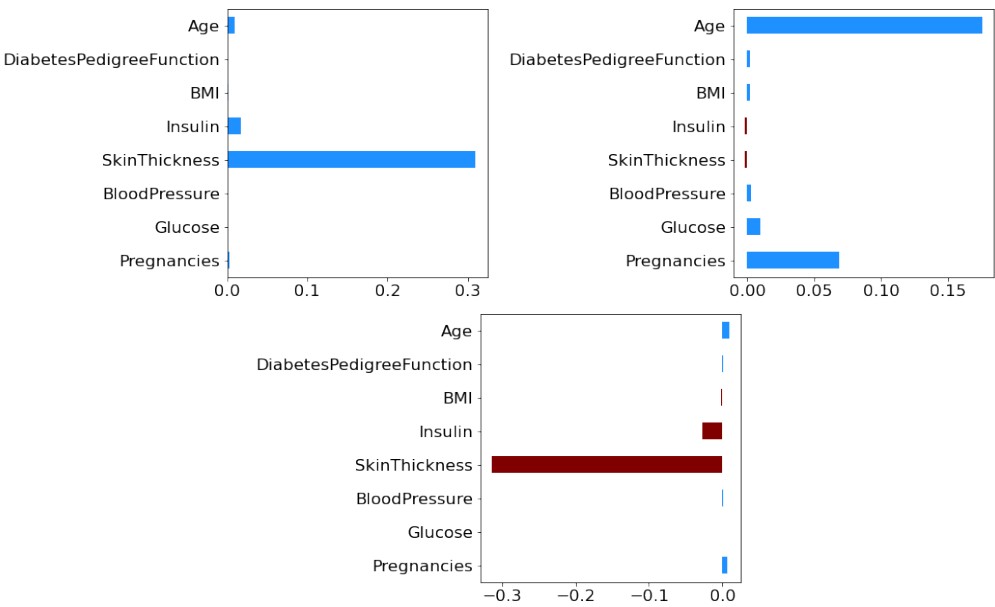

Figure 11: Feature Importance by the binary classifier for the Pima Indians Diabetes Database. a) (top-left): Feature importance for the classifier between groups 0 and 1. b) (top-right): Feature importance for the classifier between groups 0 and 2. c) (bottom-left): Feature importance for the classifier between groups 1 and 2. We note that the classifiers give significant feature importances to the features which correspond to the deltas (refer fig. 10).

