# OpenReview forum: "Explaining Low Dimensional Representation, a reproduction"
_ML_Reproducibility_Challenge/2020 — RC2020_

### Official Review · AnonReviewer3 · 2021-02-21
**Explaining Low Dimensional Representation, a reproduction**

**Rating:** 9
**Confidence:** 4

**Review:**

This report reproduces the paper "Explaining Low dimensional Representation" [1]. A detailed reproducibility summary is provided summarizing the results obtained using the implementation based on the codebase of the original authors along with a new PyTorch implementation from the efforts of this reproduction report authors. Communication with original authors (how to choose the ε hyper-parameter), hyperparameter search, discussion on results, recommendations for reproducibility and results beyond the paper (replicated the algorithm in PyTorch, new datasets and the use of more complex transformations to explain differences between clusters) are also properly constructed. Overall, this reproduction is well-constructed. Possible stretches could be the discussion of the evaluation metrics being used with respect to their suitabilities, additional metrics for a more comprehensive report and providing recommendations to the original authors for improving reproducibility.



[1] Gregory Plumb, Jonathan Terhorst, Sriram Sankararaman, and Ameet Talwalkar. Explaining groups of points in low-dimensional representations, 2020.


**Familiar With The Original Paper:**

I have read the original paper

**Reproducibility Summary:**

Report has summary

---

### Official Review · AnonReviewer1 · 2021-02-28
**We have some concerns**

**Rating:** 5
**Confidence:** 4

**Review:**

This paper checks the original paper's four claims (listed in lines 97--103) on the algorithm TGT (Transitive Global Translations). As pointed out (lines 105 - 107) by the authors, the original paper's claims are investigated via the original paper's original code. The authors of this report use new code instead to verify some extended experiments. We doubt whether this is a well-focused reproducibility investigation.

We list our concerns below.

1. Language. There are quite a few grammar problems, and the paper appears carelessly composed.

2. In the 'methodology' paragraph, by putting 'We also replicate their findings by re-implementing the authors' method in PyTorch...' does it mean that all or only a part of the computing in the original paper is re-implemented?

3. Given that the communication with the original authors does not yield meaningful result and is only because of misunderstanding, why is that paragraph still listed?

4. Could the problem with the deprecated software package be solved by Docker or Anaconda? Note that any software package shall be deprecated at some time point in the future.

5. For the clustering's uncertainty, the authors reported in the 'What was difficult' paragraph, is it because of the random seed or different implementation techniques? Is the final result sensitive to different initialisation of random clustering? It seems to us that the paper only reports the observation but has not paid any effort to uncover the cause.

6. What does 'GCE' stand for? (lines 80 & 83)

**Familiar With The Original Paper:**

I have not read the original paper

**Reproducibility Summary:**

Report has summary

---

### Official Review · AnonReviewer2 · 2021-03-02
**Rather good in content but can be improved in clarity**

**Rating:** 5
**Confidence:** 4

**Review:**

The report is on reproducing the paper "Explaining Low dimensional Representation" by Plumb et al. Overall, the submission does rather thorough reproduction and even go beyond the original paper to add some extension (though missing parts such as hyperparameter search). However, writing of the report can be largely improved as currently it looks more like a student's report rather than a work that can be published in the special issue of a journal.

Reproducibility Summary: is provided and adequately reports the major finding of the submission

Scope of reproducibility: is clearly stated and followed later

Code: the authors of this submission both used the code provided by the original authors and then re-implemented it from scratch. The github link to the code gives an error (potentially because the repository hasn't been made public yet). In anyway there was no opportunity to check the code of the submission or its docs.

Communication with original authors: the reports mentions communication with original authors regarding a choice of epsilon hyperparameter, but the authors emphasised the difficulty of selecting clusters, but no mentioning whether this question was attempted to clarify with the original authors.

Hyperparameter Search: in the experiments with the original code the authors only used the same hyperparameters as in the original paper. No hyperparameter sweep has been performed.

Ablation Study: No ablation study has been done

Discussion on results: the discussion at the end is rather well done with discussions which claims from the original paper were confirmed and which were questioned during reproduction. However, presentation of the results themselves is the poorest part of the report from the presentation point of view. The results sometimes not thoroughly discussed but the text just refer to corresponding tables and figures, which in itself is not satisfactory, moreover, those tables and figures are also missing some details and can be unclear.

Recommendations for reproducibility: the only recommendation left is in terms of this selection of clusters, which seems to cause the main issue with reproducibility

Results beyond the paper: the report provides results beyond the paper when the authors investigate a more expressive transformation than considered in the original paper. Though the results show that this extension does not actually bring significant benefits it is interesting to see these results and negative results are also very useful for the community

Overall organisation and clarity: the part that can be mostly improved. As mentioned before, the results section is the poorest here lacking the good presentation.

Some particular points (mostly on organisation and clarity):
1.	Lines 5-6, “They show their method …” – the sentence is not grammatically consistent
2.	Line 43, x_s is not defined
3.	Notation x for a low-dimensional representation in Introduction is not very suitable as it is later used to denote a point in the input space in Methodology
4.	Lines 197-198, “Since the clusters in the Tensorflow…” – the sentence is not grammatically consistent
5.	Section 5.3.2 and 5.3.3, “deltas” and “logit gammas” appear without introduction (translation and scaling were not called like this before). They are being explained only later in Section 5.3.4
6.	Section 5.3.5, G is not defined and j is reused here as it denoted the group in lower dimensional space before
7.	Line 252, “Based on the reproduction…” – unfinished sentence
8.	Section 6.1, first paragraph – translation of jupyter notebook into python scripts doesn’t seem to be a problem
9.	Line 268, “This also means …” – something wrong with this sentence
10.	Section 6.2 is redundant as this idea has been discussed at the beginning
11.	Figures 5-10 require more explanation, axes are not labelled, missing legend and overall explanation what is going on. “0th dimension” does not look good in the text

Minor:
1.	Line 171, there seems to be a typo and the second x_2 should be x_3
2.	Figures 2 and 3 – make all subfigures of the same size



**Familiar With The Original Paper:**

I have not read the original paper

**Reproducibility Summary:**

Report has summary

---

### Decision · Program_Chairs · 2021-03-31

**Decision:**

Accept

**Comment:**

Selected for ReScience-C Journal Publication.

This paper provides a valuable reproduction of the original paper, and present extensive experiments. They list the hyperparameters they used for this reproduction, which is of value for anyone looking to build on the original work. They also present extensive experiments beyond the original work, based on their pytorch reimplementation, including results on four new datasets plus synthetic data. We believe this is of good value to the community.